# Targeted depletion of uterine glandular Foxa2 induces embryonic diapause in mice

**Mitsunori Matsuo**[1,2†‡], **Jia Yuan**[1,2†§], **Yeon Sun Kim**[1,2†], **Amanda Dewar**[1,2], **Hidetoshi Fujita**[3], **Sudhansu K Dey**[1,2*], **Xiaofei Sun**[1,2*]

[1]Division of Reproductive Sciences, Cincinnati Children's Hospital Medical Center, Cincinnati, United States; [2]College of Medicine, University of Cincinnati, Cincinnati, United States; [3]Department of Biomedical Engineering, Osaka Institute of Technology, Osaka, Japan

**\*For correspondence:**
sk.dey@cchmc.org (SKD);
xiaofei.sun@cchmc.org (XS)

[†]These authors contributed equally to this work

**Present address:** [‡]Department of Obstetrics and Gynecology, University of Tokyo, Tokyo, Japan; [§]The Second Hospital and Advanced Medical Research Institute, Cheeloo College of Medicine, Shandong University, Jinan, Shandong, China

**Competing interest:** The authors declare that no competing interests exist.

**Abstract** Embryonic diapause is a reproductive strategy in which embryo development and growth is temporarily arrested within the uterus to ensure the survival of neonates and mothers during unfavorable conditions. Pregnancy is reinitiated when conditions become favorable for neonatal survival. The mechanism of how the uterus enters diapause in various species remains unclear. Mice with uterine depletion of *Foxa2*, a transcription factor, are infertile. In this study, we show that dormant blastocysts are recovered from these mice on day 8 of pregnancy with persistent expression of uterine *Msx1*, a gene critical to maintaining the uterine quiescent state, suggesting that these mice enter embryonic diapause. Leukemia inhibitory factor (LIF) can resume implantation in these mice. Although estrogen is critical for implantation in progesterone-primed uterus, our current model reveals that FOXA2-independent estrogenic effects are detrimental to sustaining uterine quiescence. Interestingly, progesterone and anti-estrogen can prolong uterine quiescence in the absence of FOXA2. Although we find that *Msx1* expression persists in the uterus deficient in *Foxa2*, the complex relationship of FOXA2 with *Msx* genes and estrogen receptors remains to be explored.

## Editor's evaluation

This manuscript reports the complex interactions that take place in the uterus between the endometrium and the blastocyst during and after embryonic diapause, a period of suspended animation that occurs in some mammals including the mouse, the model used here. The authors showed that one gene, Foxa2, interacts with two other genes, Msx1 and LIF, to control the success and duration of diapause. This will be of broad interest to researchers in the field of developmental biology and reproduction. It is a carefully done study, providing new information on the complex process that is diapause in which an embryo goes into suspended animation until it receives appropriate signaling from the uterine endometrial secretions to reactivate.

## Introduction

Embryonic diapause is a reproductive strategy in which embryo development and growth is temporarily arrested within the uterus, but is reinitiated when conditions are favorable for neonatal and maternal survival (*Cha and Dey, 2014*; *Fenelon and Renfree, 2018*). During diapause, blastocyst growth, DNA synthesis, mitosis, and metabolic activity are temporarily downregulated in the uterus to achieve a quiescent state to support blastocyst survival.

The triggers for diapause vary widely across species, ranging from photoperiod, temperature, metabolic stress, lactation, or nutrition (*Lopes et al., 2004*). It is known to occur in more than 100 species spanning over seven orders. In mice, experimental ovariectomy early on the morning of day 4, before preimplantation estrogen ($E_2$) secretion, induces embryonic diapause (*Yoshinaga and Adams, 1966*); alternatively, embryonic diapause can be induced by injections of estrogen receptor (ER) antagonists on days 3 and 4 of pregnancy to neutralize $E_2$ function (*Cha et al., 2013*). Blastocyst reactivation can be rapidly initiated by a single injection of $E_2$ in an ovariectomized dormant uterus (*Yoshinaga and Adams, 1966*). Preimplantation $E_2$ secretion on day 4 morning induces leukemia inhibitory factor (LIF) to initiate implantation. Interestingly, blastocysts in $Lif^{-/-}$ females undergo diapause (*Stewart et al., 1992*). In spite of these recognized factors, the molecular mechanism which initiates embryonic diapause is still not fully understood.

FOXA2 (forkhead box protein A2), which is expressed in glandular epithelia in the mouse uterus, plays a key role in uterine gland development and implantation. Neonatal depletion of uterine *Foxa2* causes a significant reduction in the number of glands (*Jeong et al., 2010*). Female mice with uterine glandular depletion of *Foxa2* after puberty have implantation/decidualization failure due to compromised LIF induction on day 4 of pregnancy (*Kelleher et al., 2017*).

In this study, we show that depletion of *Foxa2* in mouse uterine glands causes embryonic diapause. Dormant embryos were retrieved from uteri on day 8 of pregnancy. *Msx1* expression, which appears to be critical to maintain a quiescent uterine environment (*Cha et al., 2013*), was maintained in *Foxa2* deficient mice in our studies. Although implantation was triggered by a single LIF injection on day 8 of pregnancy in *Foxa2* deficient mice, these mice are not able to support pregnancy to full term. Furthermore, we found that balancing of estrogenic effects by either progesterone ($P_4$) supplement or application of an ER antagonist significantly improves survival of embryos in *Foxa2* deficient mice. Our study reveals that *Foxa2* plays an important role in mammalian embryonic diapause and that FOXA2-independent $E_2$ effects are detrimental to uterine quiescence during diapause.

## Results

### Uterine depletion of Foxa2 results in female infertility due to disrupted Lif induction during implantation

Previous reports have shown that *Foxa2* is expressed in the glandular epithelium before and during pregnancy (*Jeong et al., 2010*; *Wang et al., 2018*). Uterine glandular *Foxa2* is critical to normal implantation (*Kelleher et al., 2017*). Using mice with uterine-specific ($Foxa2^{f/f}Pgr^{Cre/+}$) and uterine epithelial-specific depletion of *Foxa2* ($Foxa2^{f/f}Ltf^{Cre/+}$), we confirmed that these mice are infertile (*Figure 1a*) due to the lack of *Lif* induction on day 4 of pregnancy (*Figure 1d*). Since the timing and domain of Cre activity driven by an *Ltf* or *Pgr* promoter differs (*Figure 1b*), $Foxa2^{f/f}Ltf^{Cre/+}$ females maintain FOXA2 expression in uterine glands before puberty (*Figure 1c*), whereas $Foxa2^{f/f}Pgr^{Cre/+}$ females have a minimal number of glands on postnatal day 30 (*Figure 1c*) due to early depletion of *Foxa2* (*Figure 1b*). In mature pregnant females, no positive signals of FOXA2 are observed in either of the two mouse models on day 4 of pregnancy (*Figure 1c*). The efficient depletion of *Foxa2* RNA in $Foxa2^{f/f}Ltf^{Cre/+}$ and $Foxa2^{f/f}Pgr^{Cre/+}$uteri is confirmed by quantitative PCR (*Figure 1—figure supplement 1*).

### Blastocysts enter embryonic diapause in $Foxa2^{f/f}Ltf^{Cre/+}$ and $Foxa2^{f/f}Pgr^{Cre/+}$ mice

Suppression of preimplantation $E_2$ secretion on day 4 of pregnancy renders mouse uteri quiescent to implantation. This uterine status can be extended by a daily supplement of $P_4$. Unimplanted blastocyst development in quiescent uteri is arrested (*Renfree and Shaw, 2000*; *Fenelon et al., 2014*; *Sherman and Barlow, 1972*) in $Lif^{-/-}$ females, blastocysts recovered on day 7 of pregnancy retained implantation capabilities once transferred to wild-type surrogate uteri (*Stewart et al., 1992*). This result suggests that $Lif^{-/-}$ uteri are able to maintain the quiescent phase in spite of presumed $E_2$ secretion on day 4 of pregnancy. Given the absence of LIF induction in *Foxa2* deficient uteri, we next examined whether $Foxa2^{f/f}Ltf^{Cre/+}$ and $Foxa2^{f/f}Pgr^{Cre/+}$ uteri enter diapause after day 4 of pregnancy.

Day 8 uteri were examined for implantation in $Foxa2^{f/f}Ltf^{Cre/+}$, $Foxa2^{f/f}Pgr^{Cre/+}$, and control ($Foxa2^{f/f}$) females. $Foxa2^{f/f}$ uteri show implantation sites with apparently normal morphology (*Figure 2a*). In

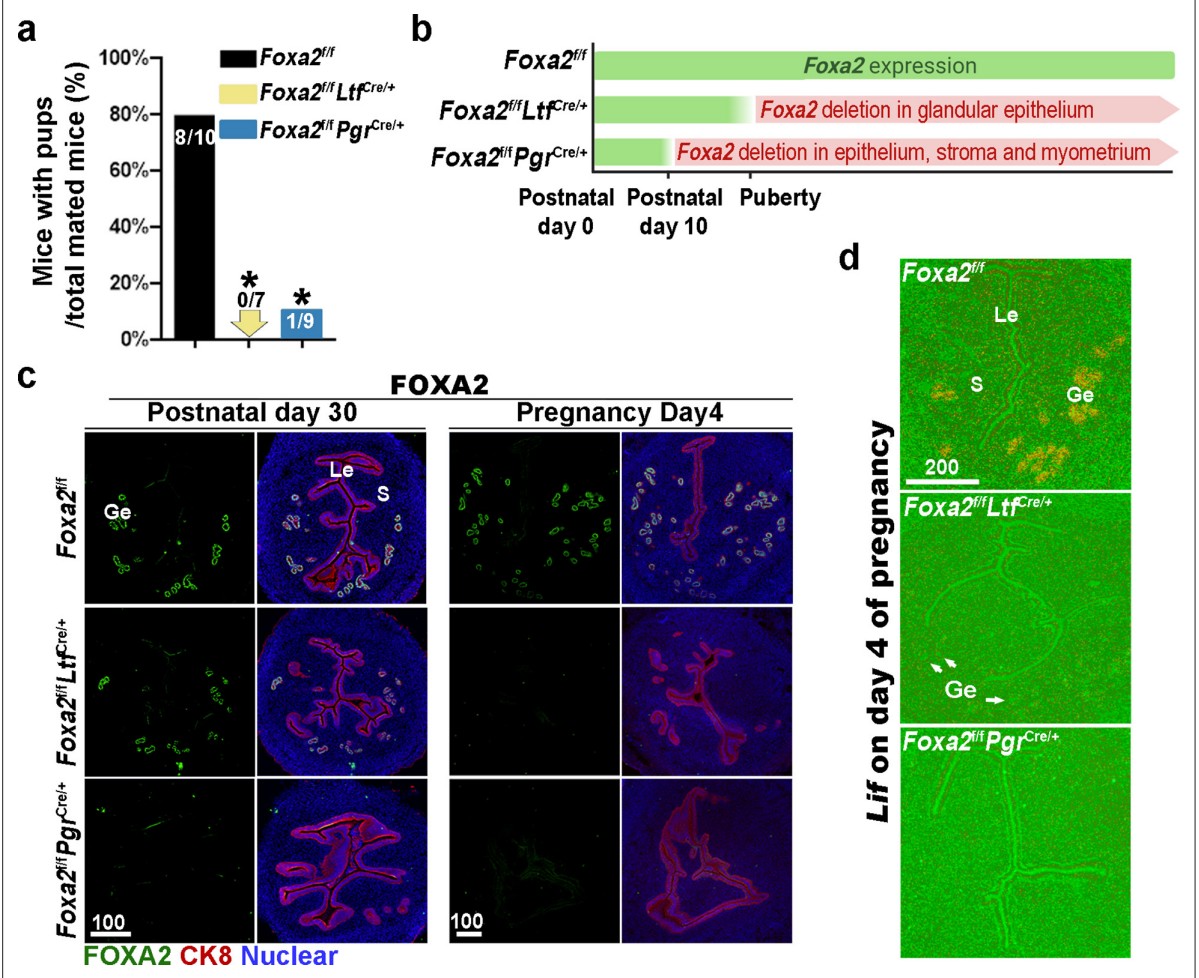

**Figure 1.** Uterine conditional depletion of *Foxa2* causes loss of leukemia inhibitory factor (LIF) secretion and female infertility. (**a**) Percentage of mice with pups per total mated mice. Numbers on bars indicate mice with pups over total number of mated mice. *p < 0.05 by Student's t-test. (**b**) Cre recombinase activity starts differently in *Foxa2^f/f^Ltf^Cre+^* and *Foxa2^f/f^Pgr^Cre+^* females. (**c**) Immunostaining of FOXA2 in the uteri on postnatal day 30 and day 4 pregnancy of *Foxa2^f/f^*, *Foxa2^f/f^Ltf^Cre+^*, and *Foxa2^f/f^Pgr^Cre+^* females. Epithelial cells are outlined by cytokeratin 8 (CK8) staining. Scale bars: 100 μm. (**d**) In situ hybridization of *Lif* in day 4 of pregnant uteri from *Foxa2^f/f^*, *Foxa2^f/f^Ltf^Cre+^*, and *Foxa2^f/f^Pgr^Cre+^* females. White arrows point to uterine glands. Scale bar: 200 μm. Le, luminal epithelia; Ge, glandular epithelia; S, stroma.

The online version of this article includes the following figure supplement(s) for figure 1:

**Figure supplement 1.** Quantitative PCR of *Foxa2* in *Foxa2^f/f^*, *Foxa2^f/f^Ltf^Cre+^*, and *Foxa2^f/f^Pgr^Cre+^* uteri tissues on day 4 of pregnancy.

contrast, the numbers of implantation sites were significantly lower in *Foxa2^f/f^Ltf^Cre/+^* or *Foxa2^f/f^Pgr^Cre/+^* females as compared to *Foxa2^f/f^* females (*Figure 2a* and *Supplementary file 1*, Table 1). The frequency of mated females with implantation sites in *Foxa2^f/f^Ltf^Cre/+^* and *Foxa2^f/f^Pgr^Cre/+^* mice was 16.7% (1/6) and 42.9% (3/7), respectively, significantly lower than those in *Foxa2^f/f^* mice (100%, 6/6) (*Supplementary file 1*, Table 1).

Dormant blastocysts were recovered by flushing *Foxa2^f/f^Ltf^Cre/+^* and *Foxa2^f/f^Pgr^Cre/+^* uterine horns on day 8 of pregnancy (*Figure 2b*). A diapause model created by ovariectomy on day 4 of pregnancy was used as a prototypical control (*Figure 2—figure supplement 1*), in which ~4.5 dormant blastocysts were retrieved. A similar number of blastocysts was recovered from *Foxa2^f/f^Ltf^Cre/+^* and *Foxa2^f/f^Pgr^Cre/+^* uterine horns on day 8 of pregnancy (*Figure 2c*), and they morphologically resembled the dormant blastocysts from the ovariectomy delayed model (*Figure 2b*). It is known that dormant blastocysts cease mitotic activity and cell proliferation (*Cha et al., 2013*). Using DAPI staining (*Figure 2d*), we found that blastocysts retrieved from *Foxa2^f/f^Ltf^Cre/+^* and *Foxa2^f/f^Pgr^Cre/+^* uteri have comparable cell numbers to those recovered from diapausing uteri achieved by ovariectomy (*Figure 2e*).

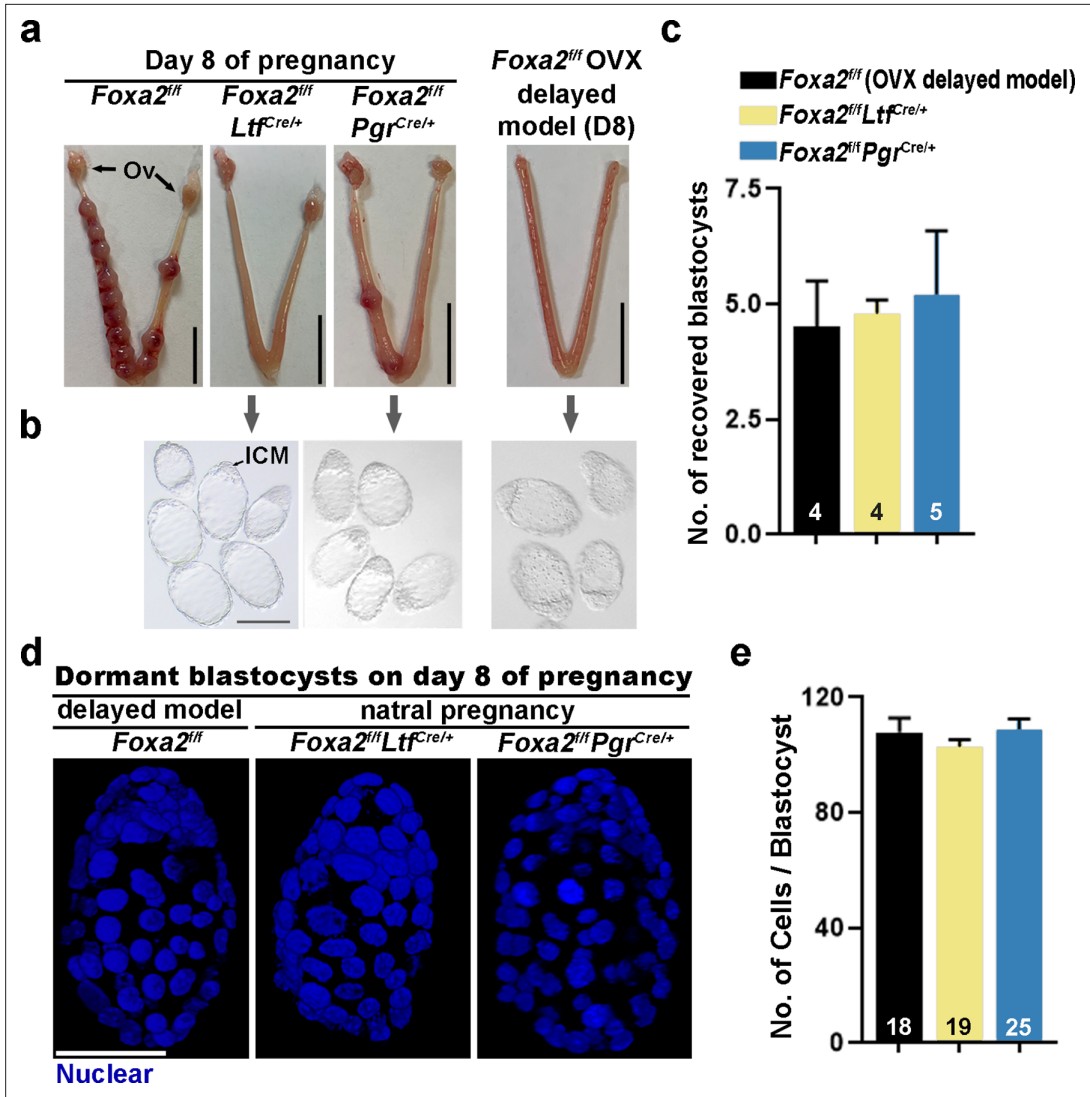

**Figure 2.** Dormant blastocysts are present in *Foxa2^f/f^Ltf^Cre+^* and *Foxa2^f/f^Pgr^Cre+^*uteri on day 8 pregnancy. (**a**) Representative photographs of day 8 pregnant uteri from *Foxa2^f/f^*, *Foxa2^f/f^Ltf^Cre+^*, and *Foxa2^f/f^Pgr^Cre+^* females. An ovariectomy-induced delayed model of *Foxa2^f/f^* mice served as a prototypical control in maintaining dormant blastocysts. Scale bar: 10 mm. Ov, ovary. (**b**) Blastocysts recovered from *Foxa2^f/f^Ltf^Cre+^* and *Foxa2^f/f^Pgr^Cre+^*uteri on day 8. Blastocysts retrieved from ovariectomized *Foxa2^f/f^* mice in delay served as controls. ICM, inner cell mass. Scale bar: 100 µm. Quantification of blastocyst numbers were shown in panel **c**. Numbers on bars indicate numbers of animals examined. Values are expressed as mean + SEM. (**d**) Representative photographs of nuclear staining of dormant blastocysts recovered from mice without implantation sites. Scale bar: 50 µm. (**e**) Average cell numbers per blastocyst. Numbers of embryos examined are shown on bars. Values are expressed as mean + SEM.

The online version of this article includes the following figure supplement(s) for figure 2:

**Figure supplement 1.** A schematic outline of sample collection from ovariectomy-induced delayed model.

## *Foxa2^f/f^Ltf^Cre/+^* and *Foxa2^f/f^Pgr^Cre/+^* uteri show characteristics of uterine quiescence

Uterine quiescence apparently depends on the presence of muscle segment homeobox (*Msx*) genes. In mice and other diapausing animals, such as in mink and Tamar Wallaby, *Msx1* and *Msx2* genes persist during diapause, but their levels are quickly suppressed with blastocyst reactivation and implantation (*Cha et al., 2013*). However, mice with uterine conditional depletion of both *Msx1* and *Msx2* fail to achieve diapause and reactivation (*Cha et al., 2013*; *Cha et al., 2020*). Since dormant blastocysts were recovered from *Foxa2^f/f^Ltf^Cre/+^* and *Foxa2^f/f^Pgr^Cre/+^* uteri on day 8 of pregnancy, we suspected that *Foxa2^f/f^Ltf^Cre/+^* and *Foxa2^f/f^Pgr^Cre/+^* uteri remain quiescent in the absence of LIF induction. We examined *Msx1* expression in the uterus on days 4 and 8 of pregnancy by fluorescence in situ hybridization.

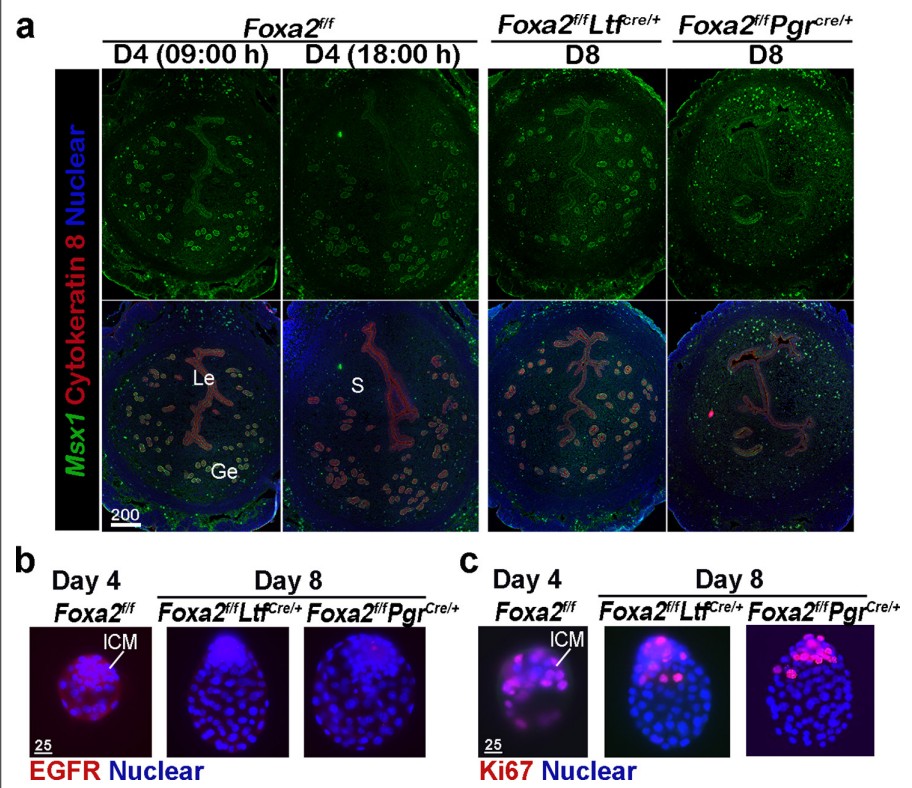

**Figure 3.** *Foxa2^{f/f}Ltf^{Cre/+}* and *Foxa2^{f/f}Pgr^{Cre/+}* females maintain uterine quiescence when examined on day 8 of pregnancy. (**a**) Fluorescence in situ hybridization of *Msx1* in days 4 and 8 pregnant uteri from *Foxa2^{f/f}*, *Foxa2^{f/f}Ltf^{Cre+}*, and *Foxa2^{f/f}Pgr^{Cre+}* females. Scale bar: 200 µm. (**b**) Epidermal growth factor receptor (EGFR) immunostaining on dormant blastocysts. Positive signals were observed in activated blastocysts recovered from *Foxa2^{f/f}* uteri on day 4 of pregnancy. Scale bar: 25 µm. (**c**) Ki67 immunostaining on dormant blastocysts collected from day 8 *Foxa2^{f/f}Ltf^{Cre+}* and *Foxa2^{f/f}Pgr^{Cre+}* females. Scale bar: 25 µm. ICM, inner cell mass.

*Msx1* signals were observed in epithelial cells before the $E_2$ surge on day 4 in *Foxa2^{f/f}* uteri, whereas luminal epithelial *Msx1* signals were suppressed after the $E_2$ secretion (*Figure 3a*; *Daikoku et al., 2011*). Remarkably, *Msx1* expression persisted in *Foxa2^{f/f}Ltf^{Cre/+}* and *Foxa2^{f/f}Pgr^{Cre/+}* luminal epithelial cells on day 8 of pregnancy (*Figure 3a*).

Epidermal growth factor receptor (EGFR) is present in day 4 blastocysts, but becomes suppressed during dormancy (*Paria et al., 1993a*). This is consistent with our current findings that EGFR expression is significantly lower in blastocysts recovered from *Foxa2^{f/f}Ltf^{Cre/+}* and *Foxa2^{f/f}Pgr^{Cre/+}* uteri on day 8 of pregnancy as compared to those retrieved from *Foxa2^{f/f}* uteri in the evening of day 4 (*Figure 3b*). The mitotic activity (Ki67 staining) in the trophectoderm of the recovered blastocysts in *Foxa2^{f/f}Ltf^{Cre/+}* and *Foxa2^{f/f}Pgr^{Cre/+}* mice on day 8 is also arrested (*Figure 3c*). This result is consistent with a study by Fujimori's group (*Kamemizu and Fujimori, 2019*). Collectively, the results suggest that *Foxa2^{f/f}Ltf^{Cre/+}* and *Foxa2^{f/f}Pgr^{Cre/+}* uteri remain quiescent until at least day 8 of pregnancy, providing a uterine environment suitable for embryonic diapause.

## Uterine activation in *Foxa2^{f/f}Ltf^{Cre/+}* and *Foxa2^{f/f}Pgr^{Cre/+}* mice deteriorates during diapause

Embryonic diapause and uterine quiescence are reversible with a single injection of $E_2$ or LIF in mice (*Yoshinaga and Adams, 1966*; *Chen et al., 2000*). Implantation failure has been shown to be rescued in *Foxa2^{f/f}Ltf^{Cre/+}* and *Foxa2^{f/f}Pgr^{Cre/+}* females by LIF administration on day 4 (*Kelleher et al., 2017*). No further analysis was carried out. Since dormant blastocysts are recovered from *Foxa2^{f/f}Ltf^{Cre/+}* and *Foxa2^{f/f}Pgr^{Cre/+}* uteri on day 8 of pregnancy in our studies, we examined if the diapausing blastocysts can be rejuvenated. *Foxa2^{f/f}Ltf^{Cre/+}* and *Foxa2^{f/f}Pgr^{Cre/+}* females received one injection of recombinant LIF (20 µg/mouse) on day 4 or 8, and the uteri were examined 2 days later (*Figure 4a*).

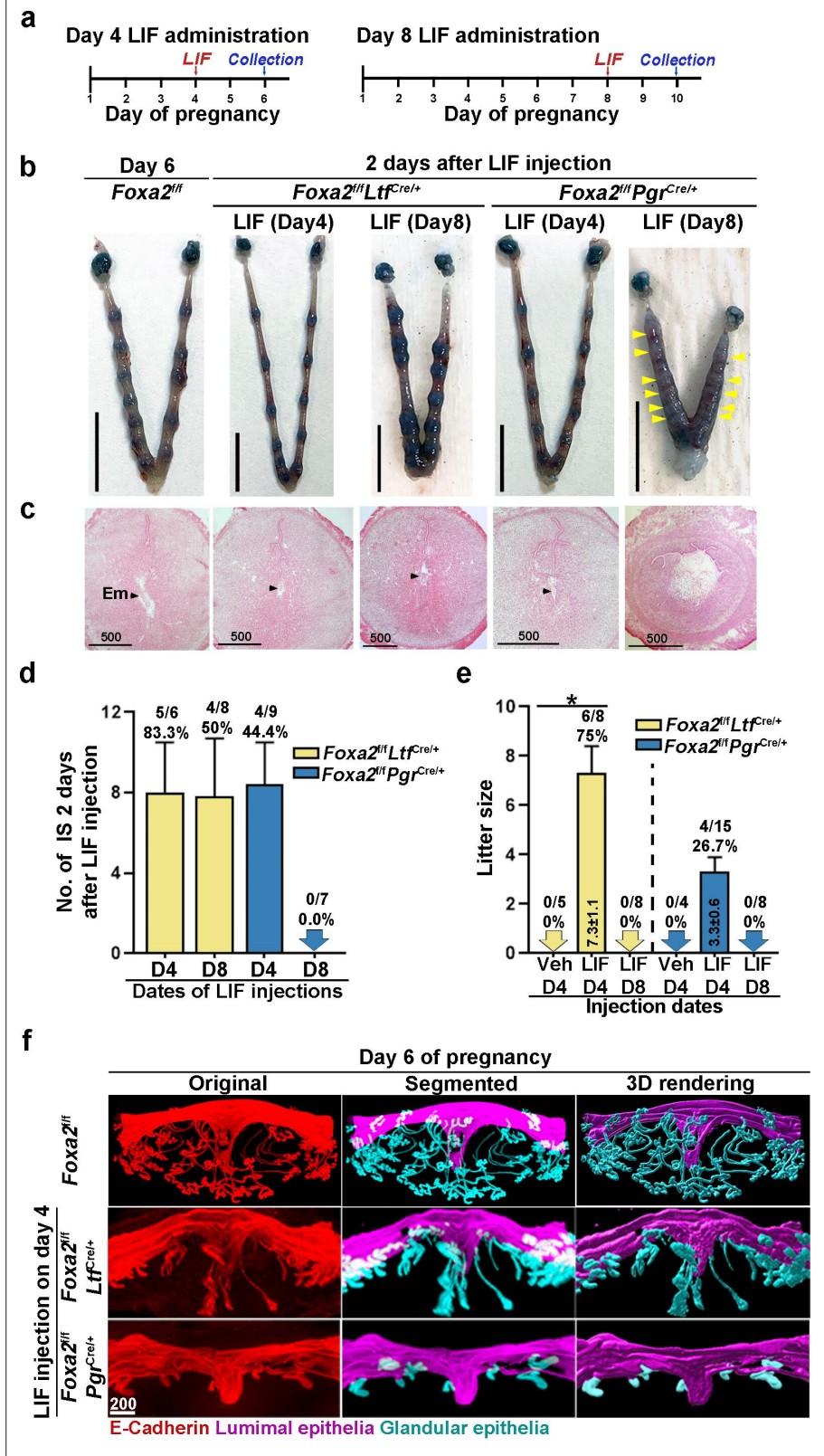

**Figure 4.** Pregnancy in *Foxa2^f/f^Ltf^Cre+^* and *Foxa2^f/f^Pgr^Cre+^* females with leukemia inhibitory factor (LIF) treatment. (**a**) Schematic outline of sample collection. LIF, LIF administration (20 μg). (**b**) Representative photograph of uteri from *Foxa2^f/f^Ltf^Cre+^* and *Foxa2^f/f^Pgr^Cre+^* females (days 6 and 10) with LIF treatment. *Foxa2^f/f^* uteri on day 6 serve as control. Scale bar: 10 mm. Histological pictures of implantation sites in panel b were presented in panel **c**. Arrowheads

*Figure 4 continued on next page*

*Figure 4 continued*

point to embryos. Em, embryo. Scale bar: 500 µm. (**d**) Average number of implantation sites in
*Foxa2^{f/f}Ltf^{Cre+}* and *Foxa2^{f/f}Pgr^{Cre+}* mice treated with LIF (20 µg) on day 4 or 8 of pregnancy. Numbers and percentage
on bars indicate mice with implantation sites over total number of mated mice. (**e**) Litter sizes of *Foxa2^{f/f}Ltf^{Cre+}* and
*Foxa2^{f/f}Pgr^{Cre+}* mice treated with LIF (20 µg) or vehicle on days 4 or 8 of pregnancy. Numbers and percentage on
bars indicate mice with pups over total number of mated mice. *p<0.05. (**f**) 3D visualization of day 6 implantation
sites in *Foxa2^{f/f}*, *Foxa2^{f/f}Ltf^{Cre+}*, and *Foxa2^{f/f}Pgr^{Cre+}* females. Images of E-cadherin immunostaining, segmented, and
3D rendered images of day 6 implantation sites in each genotype show defects in *Foxa2^{f/f}Pgr^{Cre+}* females with a LIF
injection on day 4 of pregnancy. Scale bar: 200 µm.

Consistent with the previous report (*Kelleher et al., 2017*), implantation sites were observed in both
*Foxa2^{f/f}Ltf^{Cre/+}* and *Foxa2^{f/f}Pgr^{Cre/+}* uteri if LIF was given on day 4 of pregnancy (*Figure 4b*). Further
histological evaluations reveal that normal-looking implantation chambers formed similar to those in
*Foxa2^{f/f}* mice on day 6 of pregnancy (*Figure 4b and c*). Almost all *Foxa2^{f/f}Ltf^{Cre/+}* females (five of six)
possessed implantation sites, whereas less than half (four of nine) *Foxa2^{f/f}Pgr^{Cre/+}* females had implantation sites (*Figure 4d*), although the number of implantation sites was comparable in pregnant females.

Notably, implantation sites with a normal appearance were observed in *Foxa2^{f/f}Ltf^{Cre/+}* uteri when
LIF was given on day 8 of pregnancy (*Figure 4b*), albeit edematous uteri in *Foxa2^{f/f}Pgr^{Cre/+}* with faint
blue bands. Histology of implantation sites confirmed this observation. Implantation chambers form
in *Foxa2^{f/f}Ltf^{Cre/+}* implantation sites, but neither embryos nor implantation chambers were found in
*Foxa2^{f/f}Pgr^{Cre/+}* implantation sites (*Figure 4c*). The rate of *Foxa2^{f/f}Ltf^{Cre/+}* females with implantation
sites decreased from 83.3% to 50% in females receiving LIF injection on day 4 (*Figure 4d*). All
*Foxa2^{f/f}Pgr^{Cre/+}* females showed abnormal light blue bands without recognizable implantation chambers when examined 2 days after LIF injection, and the implantation chambers showed no further
development (*Figure 4d*).

Glands have been shown to be essential for implantation and pregnancy success (*Kelleher et al.,
2017*; *Gray et al., 2001*). FOXA2 plays a key role in mouse uterine glandular genesis, and neonatal
depletion of *Foxa2* in mouse uteri causes defects in gland development (*Jeong et al., 2010*). To
examine glands, day 6 implantation sites of *Foxa2^{f/f}Ltf^{Cre/+}* and *Foxa2^{f/f}Pgr^{Cre/+}* females with LIF injection on day 4 were stained with an E-cadherin antibody. Tridimensional images were acquired as
previously described (*Yuan et al., 2018*). The number of glands in *Foxa2^{f/f}Ltf^{Cre/+}* implantation sites
was significantly reduced compared to those in natural day 6 implantation sites of *Foxa2^{f/f}* females
(*Figure 4f*). As previously reported (*Jeong et al., 2010*), glands were rarely observed in *Foxa2^{f/f}Pgr^{Cre/+}*
implantation sites. These data suggest that an increased number of glands is not required for uterine
quiescence and embryonic diapause, but the presence of a minimal number of glands is critical for
reactivation after diapause.

In mammalian embryonic diapause, arrest of blastocyst development and uterine quiescence
are transitory. Upon reactivation, the uterine environment becomes competent to support embryo
development to term when conditions are favorable for neonatal survival. To study whether reactivated uteri in *Foxa2^{f/f}Ltf^{Cre/+}* and *Foxa2^{f/f}Pgr^{Cre/+}* females are able to support full-term pregnancy, litter
sizes were counted. Six of eight *Foxa2^{f/f}Ltf^{Cre/+}*, but only 4 of 15 *Foxa2^{f/f}Pgr^{Cre/+}* females injected with
LIF on day 4 successfully delivered live pups, and *Foxa2^{f/f}Pgr^{Cre/+}* females have reduced litter sizes
(*Figure 4e*). Notably, neither *Foxa2^{f/f}Ltf^{Cre/+}* nor *Foxa2^{f/f}Pgr^{Cre/+}* females with day 8 LIF injection were
able to support full-term pregnancy, in spite of implantation occurring 2 days after LIF injection in
*Foxa2^{f/f}Ltf^{Cre/+}* females (*Figure 4e*). These results suggest that uterine readiness for reactivation in
*Foxa2^{f/f}Ltf^{Cre/+}* and *Foxa2^{f/f}Pgr^{Cre/+}* deteriorates during diapause with preimplantation $E_2$ secretion.

## Progesterone supplement during diapause improves pregnancy outcomes in *Foxa2^{f/f}Ltf^{Cre/+}* and *Foxa2^{f/f}Pgr^{Cre/+}* females after reactivation

Progesterone is required to maintain uterine quiescence and blastocyst viability in mouse embryonic
diapause. Embryonic diapause is also experimentally induced in the mouse by ovariectomy on day 4
of pregnancy before $E_2$ secretion and maintained by daily $P_4$ injections (*Yoshinaga and Adams, 1966*;
*Renfree and Fenelon, 2017*). To examine if $P_4$ levels decrease without implantation in *Foxa2^{f/f}Ltf^{Cre/+}*
and *Foxa2^{f/f}Pgr^{Cre/+}* females, we evaluated serum concentration of $P_4$ on days 4 and 8 of pregnancy

and E₂ on day 4 pregnancy in *Foxa2^{f/f}Ltf^{Cre/+}*, *Foxa2^{f/f}Pgr^{Cre/+}*, and *Foxa2^{f/f}* females. $P_4$ and $E_2$ levels in *Foxa2^{f/f}Ltf^{Cre/+}* and *Foxa2^{f/f}Pgr^{Cre/+}* females were comparable to those in *Foxa2^{f/f}* mice (**Figure 5—figure supplement 1**).

Although *Foxa2^{f/f}Ltf^{Cre/+}* and *Foxa2^{f/f}Pgr^{Cre/+}* females have normal $P_4$ and $E_2$ levels, the uterine edema in *Foxa2^{f/f}Pgr^{Cre/+}* females 2 days after LIF injection on day 8 (**Figure 4b**) suggests increased estrogenic effects during diapause. Therefore, we administered $P_4$ on days 5, 7, and 9 with LIF injection on day 8 to counter the increased estrogenic effects in *Foxa2^{f/f}Ltf^{Cre/+}* and *Foxa2^{f/f}Pgr^{Cre/+}* females (**Figure 5a**). Embryo implantation was evaluated 2 days after LIF administration. All *Foxa2^{f/f}Ltf^{Cre/+}* and *Foxa2^{f/f}Pgr^{Cre/+}* females had implantation sites with distinct blue bands (**Figure 5b and d**). Histological analysis identified embryos in the implantation chambers in mice of both genotypes (**Figure 5c**). Implantation rates and the numbers of implantation sites appear normal (**Figure 5d**). A comparable decidual response as revealed by *Bmp2* RNA levels is observed between these $P_4$ supplemented *Foxa2^{f/f}Ltf^{Cre/+}* females 2 days after LIF injection and *Foxa2^{f/f}* implantation sites on day 6 of natural pregnancy (**Figure 5—figure supplement 2**). Furthermore, around 40% of *Foxa2^{f/f}Ltf^{Cre/+}* and *Foxa2^{f/f}Pgr^{Cre/+}* females successfully delivered progeny, although litter sizes were small (2~3 pups/litter) (**Figure 5e**). These data suggest that $P_4$ supplementation improves uterine conditions during diapause in *Foxa2^{f/f}Ltf^{Cre/+}* and *Foxa2^{f/f}Pgr^{Cre/+}* mice.

## Suppression of estrogen action during diapause improves pregnancy outcomes in Foxa2^{f/f}Ltf^{Cre/+} and Foxa2^{f/f}Pgr^{Cre/+} females after reactivation

In mice, preimplantation $E_2$ secretion on day 4 of pregnancy triggers a receptive phase followed by a uterine refractory phase on day 5 onward if implantation fails to occur. This refractory phase persists until $P_4$ treatment is withdrawn (**Wang et al., 2004**). This activity suggests that $E_2$ has a biphasic effect on embryo implantation: a positive effect to induce uterine receptivity and a negative effect in changing the receptive uterus to a nonreceptive state. In *Foxa2^{f/f}Ltf^{Cre/+}* and *Foxa2^{f/f}Pgr^{Cre/+}* females, although $E_2$-induced LIF expression was abolished, FOXA2-independent negative estrogenic effects may gradually induce the refractory phase in uteri. To test this possibility, we administrated an ER antagonist (ICI-182780, named ICI) on days 3, 5, and 7 before day 8 LIF

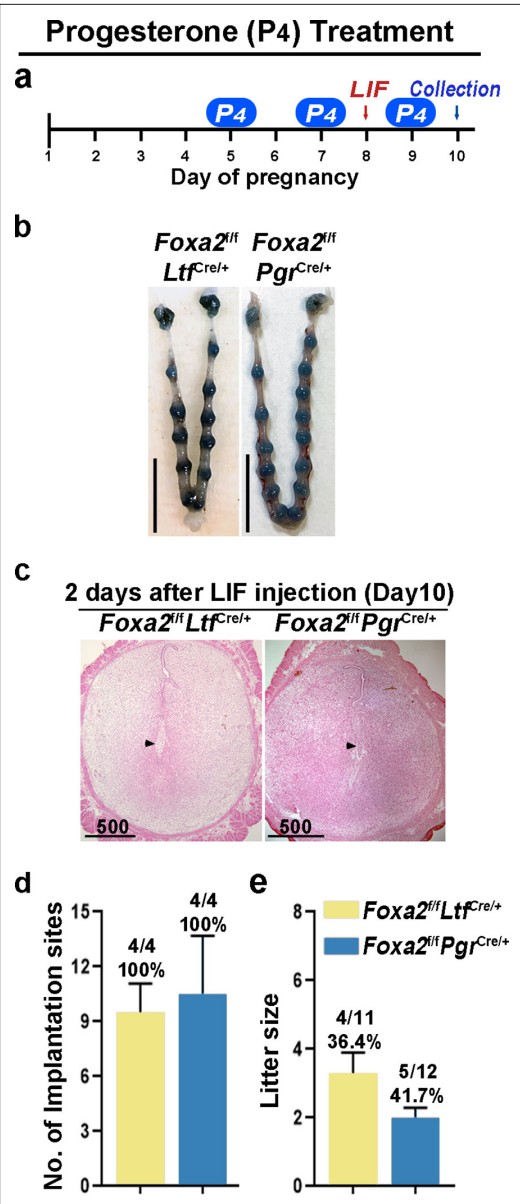

**Figure 5.** Counterbalance of estrogenic effects by $P_4$ improves maintenance of diapause in *Foxa2^{f/f}Ltf^{Cre+}* and *Foxa2^{f/f}Pgr^{Cre+}* mice. (**a**) Scheme of $P_4$ treatment. *Foxa2^{f/f}Ltf^{Cre+}* and *Foxa2^{f/f}Pgr^{Cre+}* mice were treated with leukemia inhibitory factor (LIF) (20 μg) on day 8. Pregnancy was evaluated on day 10, 2 days after LIF administration. (**b**) Representative photographs of uteri in *Foxa2^{f/f}Ltf^{Cre+}* and *Foxa2^{f/f}Pgr^{Cre+}* mice with $P_4$ supplement 2 days after LIF administration. Scale bar: 10 mm. Histological pictures of implantation sites in panel b were presented in panel **c**. Scale bar: 500 μm. (**d**) Average number of implantation sites in *Foxa2^{f/f}Ltf^{Cre+}* and *Foxa2^{f/f}Pgr^{Cre+}* mice with $P_4$ supplement. Numbers and percentage on bars indicate mice with implantation sites over total number of mated mice. (**e**) Litter sizes of *Foxa2^{f/f}Ltf^{Cre+}* and *Foxa2^{f/f}Pgr^{Cre+}* mice with $P_4$ supplement. Numbers and percentage on bars

*Figure 5 continued on next page*

*Figure 5 continued*

indicate mice with pups over total number of mated mice.

The online version of this article includes the following figure supplement(s) for figure 5:

**Figure supplement 1.** Serums levels of $P_4$ and $E_2$ in *Foxa2^{f/f}*, *Foxa2^{f/f}Ltf^{Cre+}*, and *Foxa2^{f/f}Pgr^{Cre+}*females on days 4 and 8 of pregnancy.

**Figure supplement 2.** Fluorescence in situ hybridization of *Bmp2*.

injection in *Foxa2^{f/f}Ltf^{Cre/+}* and *Foxa2^{f/f}Pgr^{Cre/+}* females (*Figure 6a*). Of note, embryonic diapause can also be experimentally induced in mice via 50 mg ICI injections on days 3 and 4 (*Cha et al., 2013*). We have also confirmed this observation in our present study. To avoid the suppression of $E_2$-induced LIF secretion on day 4, the dose of ICI was lowered to 25 mg per injection, the level at which implantation occurs normally in *Foxa2^{f/f}*females (*Figure 6b*).

Similar to $P_4$ supplement, ICI treatment improved uterine responses to LIF-induced reactivation of embryos in *Foxa2^{f/f}Ltf^{Cre/+}* and *Foxa2^{f/f}Pgr^{Cre/+}* females. Implantation sites with distinct blue bands were observed in *Foxa2^{f/f}Ltf^{Cre/+}* and *Foxa2^{f/f}Pgr^{Cre/+}* females 2 days after LIF injection (*Figure 6b*). Embryos were identified in implantation chambers in both *Foxa2^{f/f}Ltf^{Cre/+}* and *Foxa2^{f/f}Pgr^{Cre/+}* mice (*Figure 6c*). Quantitatively, 55.5% of *Foxa2^{f/f}Ltf^{Cre/+}* females and 37.5% of *Foxa2^{f/f}Pgr^{Cre/+}* females had implantation sites; the number of implantation sites in these mice is comparable to those in *Foxa2^{f/f}* females (*Figure 6d*). A comparable decidual response as indicated by *Bmp2* expression is observed between ICI-treated *Foxa2^{f/f}Ltf^{Cre/+}* females 2 days after LIF injection and *Foxa2^{f/f}* implantation sites on day 6 of natural pregnancy (*Figure 5—figure supplement 2*). Surprisingly, 50% of *Foxa2^{f/f}Ltf^{Cre/+}* females supported pregnancy to full-term with a litter size comparable to those of *Foxa2^{f/f}* females (*Figure 6e*). However, no delivery was observed in *Foxa2^{f/f}Pgr^{Cre/+}* females. These results suggest that a low level of ICI suppressed adverse estrogenic effects on uterine quiescence during diapause.

Diapause requires a favorable uterine environment to maintain dormant embryos. A putative idea of the $E_2$ secretion on day 4 mornings is to induce LIF and initiate the implantation process in a $P_4$-primed mouse uterus. In the present studies, we have used mouse models conditionally deficient in *Foxa2*, and demonstrate that LIF suppression is not sufficient to maintain long-term uterine quiescence like in ovariectomized mice maintained on a $P_4$ supplement. Our study reveals that $E_2$ has an adverse impact on uterine quiescence independent of FOXA2/LIF (*Figure 7*).

## Discussion

Over 130 mammalian species experience diapause. The triggers for diapause across species vary widely including sucking stimuli, photoperiod, the availability of nutrition, and so forth (*Fenelon et al., 2014*). The uterus is perhaps a determining factor for embryonic diapause in that a non-diapausing embryo undergoes dormancy in a diapausing uterus (*Ptak et al., 2012*). During diapause, mammals temporarily arrest blastocyst development and metabolic activity within the uterus. In normal pregnancy, uterine sensitivity to implantation is classified into three phases in mice: prereceptive, receptive, and nonreceptive (refractory) (*Wang and Dey, 2006*). Mouse uteri attain quiescence in diapause directly from the prereceptive phase via suppression of preimplantation $E_2$ secretion or LIF on day 4 of pregnancy (*Yoshinaga and Adams, 1966*; *Stewart et al., 1992*; *Paria et al., 1993b*; *Song et al., 2000*). However, the mechanism to induce embryonic diapause in mice is not clearly understood. In the current study, we show that depletion of uterine *Foxa2* triggers the mouse uterus to enter a quiescent status, which supports the arrest of embryonic development. A previous study showed that *Foxa2^{f/f}Ltf^{Cre/+}* females have implantation failure due to LIF suppression prior to implantation (*Kelleher et al., 2017*), which potentially explains why *Foxa2^{f/f}Ltf^{Cre/+}* uteri are quiescent, since a single injection of LIF is sufficient to initiate embryo implantation. Dormant blastocysts were recovered from *Lif^{-/-}* females on day 7 of pregnancy (*Stewart et al., 1992*).

Uterine depletion of *Foxa2* in either *Foxa2^{f/f}Ltf^{Cre/+}* or *Foxa2^{f/f}Pgr^{Cre/+}* females is not sufficient to maintain complete uterine quiescence by suppressing all uterine metabolic activities. In diapause, quiescent uteri are readily reactivated by an injection of $E_2$ or LIF. However, our current study showed that only *Foxa2^{f/f}Ltf^{Cre/+}* and *Foxa2^{f/f}Pgr^{Cre/+}* females with day 4 LIF injection delivered progeny; mice with day 8 LIF injection were unable to carry to term. Although *Msx1* persists in *Foxa2^{f/f}Ltf^{Cre/+}* or *Foxa2^{f/f}Pgr^{Cre/+}* uteri on day 8 of pregnancy, mice with day 8 LIF injections failed to continue pregnancy to full term, indicating the reactivation of the uteri had been compromised from day 4 to day 8. Compared with ovariectomy-induced diapause, *Foxa2^{f/f}Ltf^{Cre/+}* and *Foxa2^{f/f}Pgr^{Cre/+}* females still have

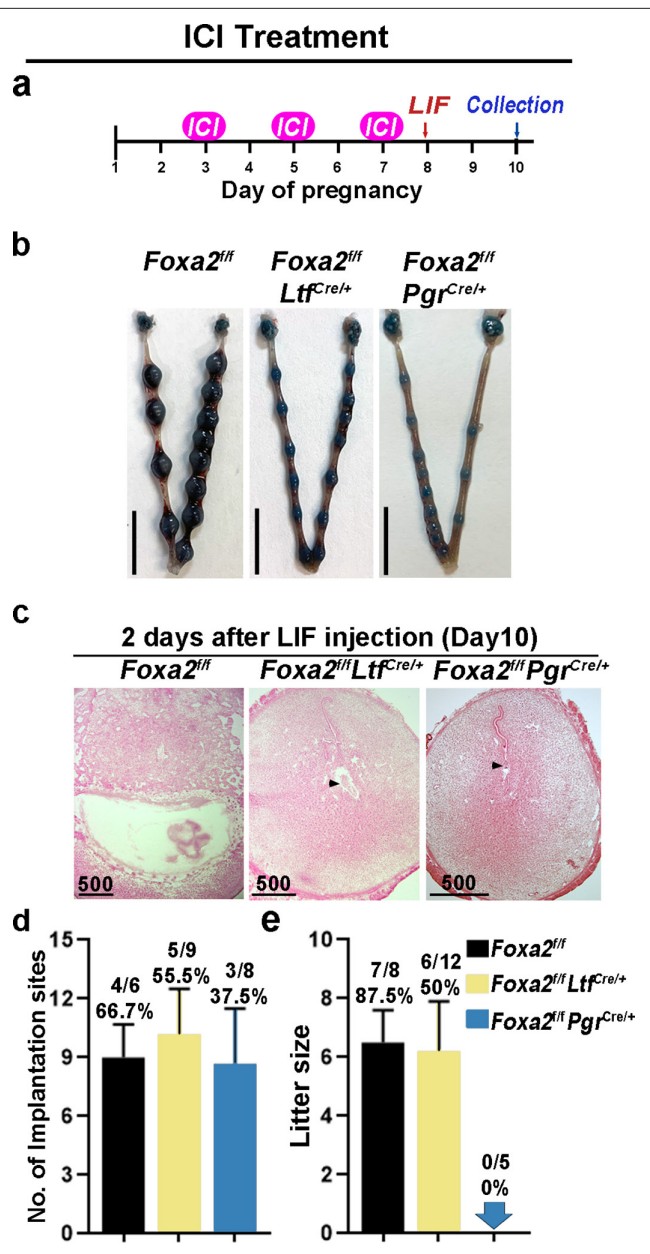

**Figure 6.** Neutralization of estrogenic effects by ICI improves diapause in *Foxa2^{f/f}Ltf^{Cre+}* and *Foxa2^{f/f}Pgr^{Cre+}* mice. (**a**) Scheme of ICI treatment. *Foxa2^{f/f}Ltf^{Cre+}* and *Foxa2^{f/f}Pgr^{Cre+}* mice were treated with leukemia inhibitory factor (LIF) (20 µg) on day 8. Pregnancy was evaluated on day 10, 2 days after LIF administration. (**b**) Representative photographs of uteri in *Foxa2^{f/f}Ltf^{Cre+}* and *Foxa2^{f/f}Pgr^{Cre+}* mice with ICI treatment 2 days after LIF administration. *Foxa2^{f/f}* mice have normal day 10 implantation sites, suggesting implantation occurs under 25 µg ICI treatment in *Foxa2^{f/f}* mice. Scale bar: 10 mm. Histological pictures of implantation sites in panel b were presented in panel **c**. Scale bar: 500 µm. (**d**) Average number of implantation sites in *Foxa2^{f/f}Ltf^{Cre+}* and *Foxa2^{f/f}Pgr^{Cre+}* mice with ICI treatment. Numbers and percentage on bars indicate mice with implantation sites over total number of mated mice. (**e**) Litter sizes of *Foxa2^{f/f}Ltf^{Cre+}* and *Foxa2^{f/f}Pgr^{Cre+}*mice with ICI treatment. Numbers and percentage on bars indicate mice with pups over total number of mated mice.

$E_2$ secretion on day 4 morning. Although $E_2$-induced LIF expression is diminished, it is possible that $E_2$ continues to have some effects, independent of FOXA2 that slowly compromises uterine readiness for reactivation in *Foxa2^{f/f}Ltf^{Cre/+}* and *Foxa2^{f/f}Pgr^{Cre/+}* females. Furthermore, we show that detrimental estrogenic effects on diapause could be countered by $P_4$ supplement or ICI treatment.

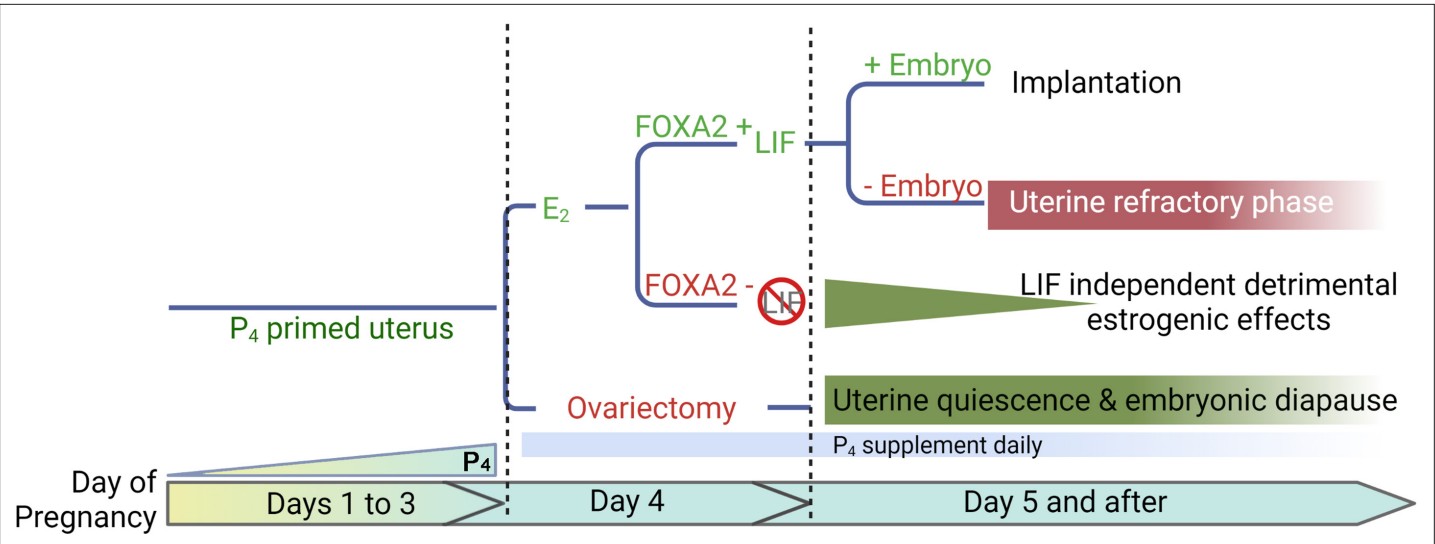

**Figure 7.** A representative scheme depicting roles of $E_2$, FOXA2 (forkhead box protein A2), and leukemia inhibitory factor (LIF) in mouse diapause. In natural pregnancy, a uterus enters a prereceptive phase before $E_2$ secretion in the morning of day 4. In the presence of FOXA2, LIF is induced by $E_2$ which renders the uterus into a receptive phase. In the event of implantation failure, the uterus becomes refractory on day 5 of pregnancy. The transition to the receptive phase is stopped if the $E_2$ secretion on day 4 is prevented by ovariectomy. The uterus remains quiescent as long as $P_4$ is supplemented; the embryonic development is arrested. A similar extension of prereceptive phase can be achieved by deleting FOXA2 or LIF. But the uterine quiescence is gradually compromised indicating that a LIF-independent estrogenic effect is detrimental to uterine quiescence.

The exact role of *Foxa2* in LIF induction by $E_2$ is not clearly understood. FOXA2 is a nuclear transcription factor and is involved in cell commitment, differentiation, and gene transcription in various organs including the lung, liver, pancreas, and gastrointestinal tract (***Lantz et al., 2004***; ***Wolfrum and Stoffel, 2006***). Since both FOXA2 and ERs are transcription factors, it is possible that FOXA2 and ER synergistically turn on LIF activity. On the other hand, ER is activated by $E_2$ secretion in the morning of day 4, whereas *Foxa2* is constantly expressed in uterine glands, suggesting an alternative possibility that FOXA2 primes transcriptional regulatory regions of *Lif*, thus enabling ER binding. In fact, previous reports showed that FOXA2 is required for chromatin opening during endoderm differentiation (***Cernilogar et al., 2019***) and for proper chromatin remodeling in human pancreas specification (***Lee, 2019***).

Estrogen is harmful to uterine quiescence in mouse diapause. Estrogen is critical for the transition from the prereceptive to the receptive phase in $P_4$-primed uteri (***Wang and Dey, 2006***). Without implantation, mouse uteri enter the refractory phase after a short receptive phase (implantation window), suggesting that $E_2$ terminates the uterine receptive phase. There is evidence that $E_2$ concentration determines the duration of the uterine receptive phase, wherein a high dose of $E_2$ shortens the receptive period (***Ma et al., 2003***). Conversely, mouse uteri in diapause are ready to be reactivated by a shot of $E_2$ as in the prereceptive phase. Ovariectomy-induced embryonic diapause could last for weeks in mice with continued $P_4$ treatment (***Ma et al., 2003***; ***Weitlauf and Greenwald, 1968***), suggesting that uterine quiescence can be maintained for significant periods of time in the absence of $E_2$. In *Foxa2$^{f/f}$Ltf$^{Cre/+}$* and *Foxa2$^{f/f}$Pgr$^{Cre/+}$* females, LIF induction is suppressed, which avoids a quick switch to the refractory phase. However, the FOXA2-independent $E_2$ effect remains in *Foxa2$^{f/f}$Ltf$^{Cre/+}$* and *Foxa2$^{f/f}$Pgr$^{Cre/+}$* uteri, slowly compromising the arrest of embryonic development and uterine quiescence. This dysfunction is further supported by our finding that $P_4$ or a low dose of ICI, which suppress $E_2$ function, improves the diapause condition in *Foxa2$^{f/f}$Ltf$^{Cre/+}$* and *Foxa2$^{f/f}$Pgr$^{Cre/+}$* females. These results indicate that estrogenic effects are not favorable to maintain diapause in mice.

## Materials and methods

### Animals and treatment

*Foxa2*[f/f] mice on a CD1 background were generated as described (*Sund et al., 2000*). This mouse line was originally obtained from Jeff Whitsett's lab at our Institute. *Foxa2*[f/f]*Ltf*[Cre+] and *Foxa2*[f/f]*Pgr*[Cre+] mice were generated by mating *Foxa2*[f/f] females with *Ltf*[Cre/+] males (C57BL/6 and albino B6 mixed background) and *Pgr*[Cre/+] mice. *Ltf*[Cre/+] and *Pgr*[Cre/+] mice on a C57BL/6 background were generated as described (*Daikoku et al., 2014*; *Soyal et al., 2005*). *Foxa2*[f/f], *Foxa2*[f/f]*Ltf*[Cre+], and *Foxa2*[f/f]*Pgr*[Cre+] mice were housed in the animal care facility at Cincinnati Children's Hospital Medical Center according to the National Institute of Health and institutional guidelines for laboratory animals. All protocols were approved by the Cincinnati Children's Animal Care and Use Committee. Mice were provided with autoclaved Laboratory Rodent Diet 5010 (Purina) and UV light-sterilized reverse osmosis/ deionized constant circulation water ad libitum. All mice used in this study were housed under a 12:12 hr light:dark cycle. At least three mice from each genotype were used for each individual experiment.

### Analysis of pregnancy events

Three adult (3 months of age) females from each genotype were randomly chosen and housed with a *Foxa2*[f/f] fertile male overnight in separate cages; the morning of finding the presence of a vaginal plug was considered successful mating (day 1 of pregnancy), and these females are designated as mated females which were selected for pregnancy experiments. For analysis of parturition, parturition events were monitored from day 18 through day 27 by observing mice daily, morning, noon, and evening.

Litter size, pregnancy rate, gestation length, and outcomes were monitored. Implantation sites were examined on pregnancy day 6 or day 8. Blue reaction was performed by intravenous injection of a blue dye solution (Chicago Blue dye) 4 min before mice were sacrificed. Distinct blue bands along the uterus indicated implantation sites. For confirmation of pregnancy in mice showing no blue bands, one uterine horn was flushed with saline and checked for the presence of blastocysts. If blastocysts were present, the contralateral horn was used for experiments; mice without any blastocysts were excluded. ICI (Fulvestrant, Sigma-Aldrich, 25 µg/mouse/day) or progesterone ($P_4$, Sigma-Aldrich, 2 mg/mouse/day) was administered in the morning (0900 hr). To induce implantation, a single injection of recombinant LIF (20 µg per mouse) was administrated in the morning (0900 hr). Embryo implantation sites were examined 2 days after LIF injection by intravenous injection of a blue dye solution.

### Histology

Tissue sections from control and experimental groups were processed on the same slide. Frozen sections (12 µm) were fixed in 4% paraformaldehyde (PFA) in PBS for 10 min at room temperature and then stained with hematoxylin and eosin (H&E) for light microscopy analysis. Images presented are representative of three independent experiments.

### Immunostaining

Staining for FOXA2 (1:300, WRAB-FOXA2, Seven Hills Bioreagents), E-cadherin (1:300, 3195s, Cell Signaling Technology), EGFR (1:100, 4267, Cell Signaling Technology), Ki67 (1:200, MA5-14520, Invitrogen), and CK8 (1:100, TROMA-1, Hybridoma Bank, Iowa) was performed using secondary antibodies conjugated with Alexa 488 or Alexa 594 (1:300, Jackson Immuno Research). Nuclear staining was performed using Hoechst 33342 (4 µg/ml, H1399, Thermo Scientific). Tissue sections from control and experimental groups were processed on the same slide for each experiment. Images presented are representative of three independent experiments.

### Whole-mount immunostaining for 3D imaging

To reveal the tridimensional visualization of implantation sites, whole-mount immunostaining with 3DISCO clearing was performed as previously described (*Yuan et al., 2018*). Anti-E-cadherin antibody (1:100, 3195s, Cell Signaling Technology) was used to stain the luminal epithelium. 3D images were acquired by a Nikon upright confocal microscope (Nikon A1R). To construct the 3D structure of the tissue, the surface tool in Imaris (Bitplane) was used.

### Fluorescence in situ hybridization

Digoxigenin (DIG)-labeled probes were generated according to the manufacturer's protocol (Roche). PFA-fixed frozen sections from control and experimental groups were hybridized with DIG-labeled

cRNA probes.Frozen sections (12 μm) from each genotype and treatment group were processed on the same slide for each probe. Briefly, following fixation (in 4% PFA/PBS) and acetylation, slides were hybridized at 55°C with DIG-labeled *Lif* and *Msx1* probe. Anti-DIG-peroxidase was applied onto hybridized slides following washing and peroxide quenching. Color was developed by TSA (Tyramide Signal Amplification) fluorescein according to the manufacturer's instructions (PerkinElmer). Nuclear staining was performed using Hoechst 33342 (4 μg/ml, H1399, Thermo Scientific). Images presented are representative of three independent experiments.

### In situ hybridization using radioactive probes

In situ hybridization using radioactive ($^{35}$S GTP) labeled *Lif* probes was performed as previously described (*Tan et al., 1999*). In brief, frozen sections (12 μm) were mounted onto poly-L-lysine-coated slides and fixed in cold 4% PFA in PBS. The sections were prehybridized and hybridized at 45°C for 4 hr in 50% (vol/vol) formamide hybridization buffer containing $^{35}$S-labeled anti-sense RNA probes (PerkinElmer). RNase A-resistant hybrids were detected by autoradiography. All sections were post-stained with H&E. Images presented are representative of three independent experiments.

### Progesterone ($P_4$) and estradiol-17b ($E_2$) assays

Mouse blood samples were collected at 9:00 am on days 4 and 8 of pregnancy. Serum was separated by centrifugation and stored at –80°C until analysis. Serum hormonal levels in the serum were measured by $P_4$ or $E_2$ EIA kit (Cayman Chemical) as previously described (*Daikoku et al., 2011*).

### Quantitative RT-PCR

RNAs from *Foxa2*$^{f/f}$, *Foxa2*$^{f/f}$*Ltf*$^{Cre+}$, and *Foxa2*$^{f/f}$*Pgr*$^{Cre+}$mice uterine samples were analyzed as described previously (*Sun et al., 2014*; *Das et al., 1995*). In brief, total RNA was extracted with Trizol (Invitrogen, Waltham, MA) according to the manufacturer's protocol. After DNase treatment (Ambion, Austin, TX), 1 μg of total RNA was reverse-transcribed with Superscript II (Invitrogen). Real-time PCR was performed using primers 5'-GACATACCGACGCAGCTACA-3' (sense) and 5'- GCCGGTAGAAAG GGAAGAGG-3' (anti-sense) for mouse *Foxa2*; 5'- GCAGATGTACCGCACTGAGATTC-3' (sense) and 5'-ACCTTTGGGCTTACTCCATTGATA-3' (anti-sense) for mouse *Rpl7*.

### Statistical analysis

Each experiment was repeated at least three times using independent samples. Data are shown as mean ± SEM. Statistical analyses were performed using a two-tailed Student's t-test. A value of $p < 0.05$ was considered statistically significant.

## Acknowledgements

We thank Katie Gerhardt for her excellent editing of the manuscript. This work was supported in parts by NIH grants (HD103475 and HD068524 to SKD). YSK is supported by a National Research Foundation of Korea (NRF) grant (NRF-2021R1A6A3A03038446).

## Additional information

### Funding

| Funder | Grant reference number | Author |
| --- | --- | --- |
| Eunice Kennedy Shriver National Institute of Child Health and Human Development | HD103475 | Sudhansu K Dey |
| Eunice Kennedy Shriver National Institute of Child Health and Human Development | HD068524 | Sudhansu K Dey |

| Funder | Grant reference number | Author |
| --- | --- | --- |
| National Research Foundation of Korea | NRF-2021R1A6A3A03038446 | Yeon Sun Kim |

The funders had no role in study design, data collection and interpretation, or the decision to submit the work for publication.

## Author contributions

Mitsunori Matsuo, Conceptualization, Data curation, Visualization, Writing – original draft; Jia Yuan, Conceptualization, Data curation, Validation, Visualization; Yeon Sun Kim, Data curation, Validation, Visualization; Amanda Dewar, Data curation, Validation; Hidetoshi Fujita, Resources, Data curation; Sudhansu K Dey, Conceptualization, Formal analysis, Supervision, Writing – review and editing; Xiaofei Sun, Conceptualization, Formal analysis, Supervision, Writing – original draft, Writing – review and editing

## Author ORCIDs

Jia Yuan http://orcid.org/0000-0002-4807-8646
Yeon Sun Kim http://orcid.org/0000-0003-4854-8334
Sudhansu K Dey http://orcid.org/0000-0001-9159-186X
Xiaofei Sun http://orcid.org/0000-0001-9601-5423

## Decision letter and Author response

Decision letter https://doi.org/10.7554/eLife.78277.sa1
Author response https://doi.org/10.7554/eLife.78277.sa2

## Additional files

### Supplementary files

• Supplementary file 1. Table 1. Implantation sites in $Foxa2^{f/f}$, $Foxa2^{f/f}Ltf^{Cre+}$, and $Foxa2^{f/f}Pgr^{Cre+}$ females on day 8 of pregnancy.

• Transparent reporting form

### Data availability

All data are included in the manuscript.

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
