## [Editor Report]

This manuscript reports the complex interactions that take place in the uterus between the endometrium and the blastocyst during and after embryonic diapause, a period of suspended animation that occurs in some mammals including the mouse, the model used here. The authors showed that one gene, Foxa2, interacts with two other genes, Msx1 and LIF, to control the success and duration of diapause. This will be of broad interest to researchers in the field of developmental biology and reproduction. It is a carefully done study, providing new information on the complex process that is diapause in which an embryo goes into suspended animation until it receives appropriate signaling from the uterine endometrial secretions to reactivate.

---

## [Decision Letter]

**Decision letter after peer review:**

[Editors’ note: the authors submitted for reconsideration following the decision after peer review. What follows is the decision letter after the first round of review.]

Thank you for submitting the paper "Targeted deletion of uterine glandular Foxa2 induces embryonic diapause in mice" for consideration by *eLife*. Your article has been reviewed by 3 peer reviewers, one of whom is a member of our Board of Reviewing Editors, and the evaluation has been overseen by a Senior Editor. The following individual involved in the review of your submission has agreed to reveal their identity: Marilyn B Renfree (Reviewer #2).

Comments to the Authors:

We are sorry to say that, after consultation with the reviewers, we have decided that this work will not be considered further for publication by *eLife*.

The manuscript is descriptive, and the findings are not particularly novel, given the known effects of Foxa2 depletion in the uterus, causing glandular defects. Foxa2 regulates LIF, and it is a well-known fact that Lif is essential for implantation. In addition, the model has not been adequately defined in terms of the depletion of the target genes. The demonstration that the embryos are in diapause is not convincing and recovered embryos should have been cultured or transplanted to demonstrate whether they are viable.

*Reviewer #1 (Recommendations for the authors):*

Molecular characterization of FOXA2 action needs to be carried out in addition to the interesting phenotype observed to make this manuscript more interesting to *eLife* readers.

*Reviewer #2 (Recommendations for the authors):*

A set of private recommendations for the authors that outline how you think the science and its presentation could be strengthened.

This is such a complex set of interactions that would be made so much clearer if the authors would provide a line diagram summarising the key results and the relationship of each of the questions tested.

Also, they need to make it clear at what stage of pregnancy the Cre recombinase effects for the two mutant mice take place. It was confusing to understand the relationship between the various different stages of development (eg before puberty, in mature pregnant females, at day 4 of pregnancy, etc) examined to the timing of administration of the treatments. A table summarising these would help.

The assumption was made that "plug-positive females' is instantly understood by everyone. Why not say mated females, or define it further in other ways.

*Reviewer #3 (Recommendations for the authors):*

1. Throughout, the Cre/lox effects should be referred to as depletion of Foxa2 rather than deletion or suppression.

2. Line 76: What does the quality of embryonic diapause mean?

3. Lines 110 et seq. The text indicates that implantation sites were rarely observed in either of the Cre/lox models, but then indicates that the frequency of mice with implantation sites was 16.7 and 42.9%. These do not seem to be rare occurrences. Supplemental Table 1 indicates that the number of implantation sites where present was within the normal range for mouse implantation in the depleted phenotypes.

4. Line 152: The concept of uterine readiness to be activated is vague and needs to be defined.

---

## [Author Response]

[Editors’ note: The authors appealed the original decision. What follows is the authors’ response to the first round of review.]

Reviewer #1 (Recommendations for the authors):Molecular characterization of FOXA2 action needs to be carried out in addition to the interesting phenotype observed to make this manuscript more interesting to eLife readers.

The current study addresses the question as to how the uterus enters into and maintains diapause. A study has shown that LIF is critical to implantation, and embryos in LIF KO mice enter diapause. However, our study shows this diapause in LIF KO lasts for a short period, whereas LIF independent estrogenic effects have adverse effects on embryonic diapause. Using mouse models deficient in Foxa2, we dissected out the LIF-dependent and -independent effects of estrogen secreted on day 4. The current study reveals previously unrealized roles of estrogen on uterine quiescence.

This reviewer’s interests regarding the exact role of Foxa2 in LIF induction is not within the scope of this study. However, this should not marginalize the importance of the present study. The interplay of transcription factors involving FoxA2, Msx1 and Esr1 in Lif induction and regulation of embryonic diapause will require further investigation for years.

Reviewer #2 (Recommendations for the authors):A set of private recommendations for the authors that outline how you think the science and its presentation could be strengthened.This is such a complex set of interactions that would be made so much clearer if the authors would provide a line diagram summarising the key results and the relationship of each of the questions tested.Also, they need to make it clear at what stage of pregnancy the Cre recombinase effects for the two mutant mice take place. It was confusing to understand the relationship between the various different stages of development (eg before puberty, in mature pregnant females, at day 4 of pregnancy, etc) examined to the timing of administration of the treatments. A table summarising these would help.The assumption was made that "plug-positive females' is instantly understood by everyone. Why not say mated females, or define it further in other ways.

We agree with these suggestions. We have now included a summarizing scheme (Figure 7). The timing of various treatments in different mouse models are shown in figures 4a, 5a and 6a.

All “plug-positive” are changed to “mated”.

Reviewer #3 (Recommendations for the authors):1. Throughout, the Cre/lox effects should be referred to as depletion of Foxa2 rather than deletion or suppression.

Manuscript has been revised accordingly.

2. Line 76: What does the quality of embryonic diapause mean?

We have revised the sentence in line 74. Notably, we found that balancing estrogenic effects by progesterone (P_4_) supplement or neutralization of estrogenic effects by an estrogen receptor antagonist significantly improves survival of embryos in *Foxa2* deficient mice.”

3. Lines 110 et seq. The text indicates that implantation sites were rarely observed in either of the Cre/lox models, but then indicates that the frequency of mice with implantation sites was 16.7 and 42.9%. These do not seem to be rare occurrences. Supplemental Table 1 indicates that the number of implantation sites where present was within the normal range for mouse implantation in the depleted phenotypes.

We revised the description in lines 110-112. In contrast, the numbers of implantation sites were significantly lower in *Foxa2^f/f^Ltf^Cre/+^* or *Foxa2^f/f^Pgr^Cre/+^* females as compared to *Foxa2^f/f^* females (Figure 2a and Supplemental Table1).

4. Line 152: The concept of uterine readiness to be activated is vague and needs to be defined.

We revised this statement in line 155